

# Screening of prognostic biomarkers for endometrial carcinoma based on a ceRNA network

Ming-Jun Zheng, Rui Gou, Wen-Chao Zhang, Xin Nie, Jing Wang, Ling-Ling Gao, Juan-Juan Liu, Xiao Li and Bei Lin

Department of Gynaecology and Obstetrics, Shengjing hospital affiliated to China Medical University, Liaoning, China
Key laboratory of Maternal-Fetal Medicine of Liaoning Province, Key laboratory of Obstetrics and Gynecology of higher education of Liaoning Province, Liaoning, China

## ABSTRACT

**Objective**. This study aims to reveal the regulation network of lncRNAs-miRNAs-mRNA in endometrial carcinoma (EC), to investigate the underlying mechanisms of EC occurrence and progression, to screen prognostic biomarkers.

**Methods**. RNA-seq and miRNA-seq data of endometrial carcinoma were downloaded from the TCGA database. Edge.R package was used to screen differentially expressed genes. A database was searched to determine differentially expressed lncRNA-miRNA and miRNA-mRNA pairs, to construct the topological network of ceRNA, and to elucidate the key RNAs that are for a prognosis of survival.

**Results**. We screened out 2632 mRNAs, 1178 lncRNAs and 189 miRNAs that were differentially expressed. The constructed ceRNA network included 97 lncRNAs, 20 miRNAs and 73 mRNAs. Analyzing network genes for associations with prognosies revealed 169 prognosis-associated RNAs, including 92 lncRNAs, 16miRNAs and 61 mRNAs.

**Conclusion**. Our results reveal new potential mechanisms underlying the carcinogenesis and progression of endometrial carcinoma.

## INTRODUCTION

Endometrial carcinoma (EC) is the most common malignant gynecologic tumor. It is also the fourth most common malignant tumor in women following breast, lung and colorectal cancers (*Siegel, Miller & Jemal, 2017*). The absence of early diagnostic and therapeutic markers are the main reasons for death caused by EC. It has been found that EC development is related not only to hormones, but also to gene mutation. EC is mainly classified into two subtypes: estrogen dependent subtype (subtype I) and gene mutation-associated subtype (subtype II). Subtype I comprises 80% of new incidence, is relatively well-differentiated, and has a high five-year survival rate. Subtype II (gene mutation-associated) EC is histologically poorly differentiated, highly invasive and has a five-year survival rate of only 30–40% (*Bokhman, 1983*; *Jia et al., 2014*; *Matias-Guiu & Davidson, 2014*). It has been shown that mutations in p53, ATR and other genes

Corresponding author
Bei Lin, linbei88@hotmail.com

correlate with EC carcinogenesis and prognosis (*Li et al., 2014*; *Zighelboim et al., 2009*). In recent years, competing endogenous RNA (ceRNA) (*Tay, Rinn & Pandolfi, 2014*) has become a hotspot in cancer research. MicroRNAs (miRNAs) play important roles in post-transcriptional regulation by inhibiting target gene expression; EC carcinogenesis is also associated with miRNA expression. It was revealed that miR-182 promotes EC cell proliferation by targeting the tumor suppressor gene TCEAL7 (*Guo et al., 2013*). *Sun et al. (2017a)* constructed the TF-miRNA-mRNA network and found that miR-141-3p and miR-130b-3p are involved in the carcinogenesis and development of EC. Long non-coding RNAs act as key players regulating gene expression during the EC transformation and progression through signaling, decoying, guiding and scaffolding at the epigenetic, transcriptional and post-transcriptional levels (*Smolle et al., 2015*; *Rinn & Chang, 2012*; *Carlson et al., 2015*; *Martianov et al., 2007*; *Yoon, Abdelmohsen & Gorospe, 2013*; *Wang & Chang, 2011*). Several lncRNAs have been found to work as prognostic indicators for cancer. Through the ceRNA regulation network in ovarian cancer, *Zhou et al. (2016)* identified a group of prognostic biomarkers for ovarian cancer comprising 10 lncRNAs. *Xu et al. (2016)* utilized lncRNA-mRNA expression profiling, finding that six lncRNAs, including KIAA0087, correlated with EC prognosis. Based on these findings, we believe that classification based on new molecular subtypes would benefit the early diagnosis and treatment of EC.

Interaction mechanisms in the lncRNA-miRNA-mRNA regulation network in EC remain unclear. Thus, investigating the relationships among these molecules and to find effective tumor biomarkers is critical to improve patient prognoses. Therefore, we constructed a topological network of ceRNA in EC and screened RNA biomarkers for prognosis, thereby identifying 92 lncRNAs, 16 miRNAs and 61mRNAs that are related to patient prognosis and survival. Finally, we defined a group of prognostic biomarkers comprising eight mRNAs.

## MATERIALS AND METHODS

### Data sources and pretreatment

EC RNA-seq and miRNA-seq data were downloaded from the TCGA database (https://cancergenome.nih.gov/) (*Tomczak, Czerwińska & Wiznerowicz, 2015*). RNA-seq data included 551 EC and 35 paracancerous samples. miRNA-seq data included 546 EC and 33 paracancerous samples. As the data was obtained from the TCGA website, further approval by an ethics committee was not required. This study meets the publication guidelines provided by TCGA (http://cancergenome.nih.gov/publications/publicationguidelines). RNA-seq and miRNA-seq matrix files describing samples were downloaded from TCGA. After that, mRNA and lncRNA expression profiles were extracted from the RNA-seq matrix information. Thus, we obtained three matrix files: mRNA, lncRNA and miRNA expression profiles.

### Screening of differentially expressed genes

Statistical analysis was performed using the exactTest function in the edge.R software package (*Robinson, McCarthy & Smyth, 2010*). and differentially expressed genes were

screened out. Next, the *P* value was FDR-corrected. mRNA, lncRNA and miRNA with FDR < 0.05 were screened out. RNAs with fold change >4 ($|\text{logFC}| > 2$) were further selected as differentially expressed mRNA, lncRNA and miRNAs. A volcano plot of differentially expressed genes was generated.

## ceRNA topological network construction
### *lncRNA-miRNA pair screening*
MiRcode (*Jeggari, Marks & Larsson, 2012*) is database that predicts miRNA targets based on a transcriptome annotated by human GENCODE. Through miRcode matching, we determined the miRNAs that interacted with differentially expressed lncRNAs and compared them to the screened-out differentially expressed miRNAs. The miRNAs shared by the two miRNA pools were selected and paired with corresponding lncRNAs to generate the final lncRNA-miRNA pairing files.

### *Screening of miRNA-mRNA pairs*
MiRDB (*Wong & Wang, 2015*) is a database that can be used to predict miRNA target genes based on high throughput sequencing data. **miRTarBase** (*Chou et al., 2016*) is an experimentally-verified miRNA target gene database. TargetScan (*Lewis, Burge & Bartel, 2005*) predicts miRNA targets by searching for the presence of conserved 8mer and 7mer sites that match the seed region of an input miRNA. We predicted target genes of the miRNA in the lncRNA-miRNA pairs through these three aforementioned databases, and defined target genes that were predicted by all three databases (miRDB, miRTarBase and TargetScan) as the final screened-out target genes. These genes were compared to the differentially expressed genes mentioned above, and genes in common were selected to build an miRNA-mRNA pairing file. Finally, a ceRNA topological network was constructed based on the abovementioned pairing files.

## Topological and stability analysis of the ceRNA network
The NetWorkAnalyzer toolkit (*Shannon et al., 2003*) from Cytoscape (*Doncheva et al., 2012*) was used to analyze the characteristics of the ceRNA topological network. NetWorkAnalyzer is mainly used to analyze network diameter, connection numbers and average clustering coefficient. In this study, we calculated network connections, the path length and the closest centrality of nodes.

## Functional and enrichment analysis of ceRNA
Biological function characterization of DEmRNAs in the ceRNA network was performed using the Database for Annotation, Visualization, and Integrated Discovery (DAVID) version 6.7 (https://david-d.ncifcrf.gov/home.jsp), and pathway enrichment analysis was performed using the KO-Based Annotation System (KOBAS) 3.0 (http://kobas.cbi.pku.edu.cn/) (*Huang da, Sherman & Lempicki, 2009*; *Xie et al., 2011*).

## Prognostic analysis of the ceRNA module
Clinical information of samples was downloaded from the TCGA database, and survival data were extracted. Combined with expression profile data, a Kaplan–Meier (K–M) survival curve was generated for each node in the ceRNA topological network using

survival (*Therneau, 2015*) in the R package. Survival differences for genes, lncRNAs and miRNAs in the ceRNA module were also analyzed. Patients were dichotomized for survival analysis using a log-rank test with optimal cutoff values determined by the "surv_cutpoint" function of the "survminer" R package. $P < 0.05$ was considered statistically significant.

### Validation of DEmRNAs and DEmiRNAs in the ceRNA network

The GSE17025 (*Allard et al., 2008*) and GSE35794 datasets were used to validate the expression of candidate mRNAs and miRNAs, respectively. Transcriptome profiling data in the GSE17025 (Platform: GPL570) dataset contained 91 EC and 12 non-tumor samples. The microRNA expression profile in the GSE35794 (Platform: GPL10850) dataset contained 18 EEC and four normal samples.

## RESULTS

### Screening of differentially expressed RNAs

The mRNA expression profiles of 35 normal samples and 551 EC samples were compared. After statistical validation, 2632 differentially expressed mRNAs were screened out including 1672 upregulated and 960 downregulated mRNAs (Figs. 1A and 1D; Table S1). We obtained 1178 differentially expressed lncRNAs after screening, including 867 that were upregulated and 311 that were downregulated. Results are shown in Figs. 1B and 1E; Table S1.

The miRNA expression profiles of 33 normal samples and 546 EC samples were compared, and 189 differentially expressed miRNAs were obtained after statistical validation. One hundred and forty of these 189 miRNAs were upregulated and 49 were downregulated. Results are shown in Figs. 1C and 1F; Table S1.

### Construction of the ceRNA network and analysis of topology and stability

Based on the differentially expressed lncRNA and miRNA profiles, lncRNA and corresponding miRNAs were paired using the miRcode database. We established 556 pairs, with 99 lncRNAs and 27 miRNAs (Table S2). Twenty-five of these 27 DEmiRNAs were found in the starBase database. The target genes of these 25 miRNAs were predicted using three public databases: miRDB, miRTarBase and TargetScan. Nine hundred and fifty-six target genes were found in total. These target genes were compared with 2632 differentially expressed genes that were screened out, and genes in common were selected out. Seventy-three DEmRNAs that could interact with 20 of the 25 DEmiRNAs in all three datasets were selected (Table S3). After removing the remaining seven DEmiRNAs and corresponding lncRNAs, 97 DElncRNAs, 20 DEmiRNAs and 73 DEmRNAs were ultimately used to construct a ceRNA network used the "**ggalluvial**" R package for visualization (Fig. 2A; Table S4).

The topology of the ceRNA network was analyzed to verify the network's reliability. The distribution of the numbers of nodes is shown in Fig. 2B. Node number decreased with increases in distribution degree. The highest degree was 51; the lowest was zero. Most nodes in the network were nodes with fewer interactions. The closeness centrality (CC) of
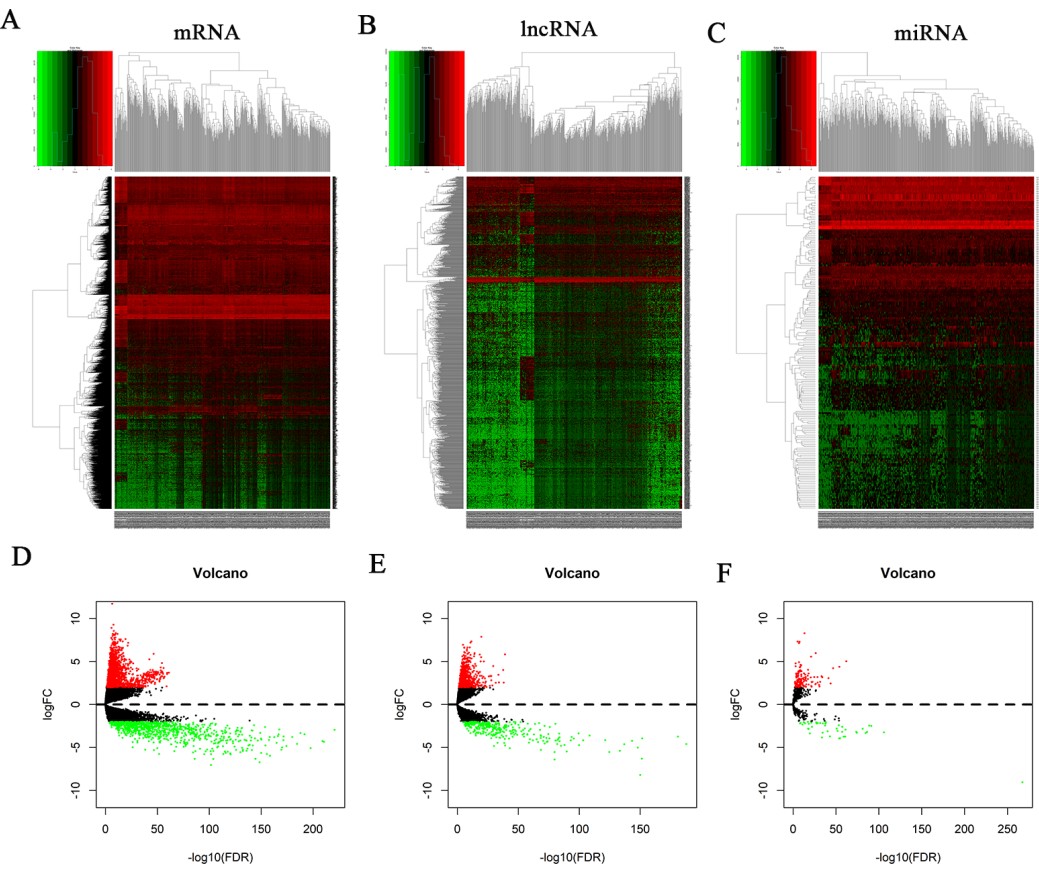

**Figure 1** **Heatmap and volcano plots of differentially expressed RNAs.** (A) Heatmap plots of differentially expressed mRNAs. (B) Heatmap plots of differentially expressed lncRNAs. (C) Heatmap plots of differentially expressed miRNAs. The horizontal axis represents samples. Sample clusters are presented above the horizontal axis. The vertical axis represents RNAs. Red denotes upregulated genes, and green denotes downregulated genes. (D) Volcano plots of differentially expressed mRNAs. (E) Volcano plots of differentially expressed lncRNAs. (F) Volcano plots of differentially expressed miRNAs. The *Y*-axis denotes the log of FC (base 2) and the *X*-axis plots the negative log of the false discovery rate (FDR; base 10). Each point represents a gene. Green dots represent downregulated RNAs, red dots represent upregulated RNAs, and black dots represent non-DEGs.

a given node can be used calculate the connection steps from the node to another. More centralized nodes show higher scores. Closeness centrality indicates the shortest path, and we demonstrate in Fig. 2C that there were more nodes displaying the same numbers of connections, and that these nodes are relatively centralized in the network. Those nodes with more connections are relatively sparsely distributed. Path reflects the combination of all nodes in the network. Figure 2D shows the shortest pathway length distribution in the network, demonstrating that path length is mainly focused in the center. There were fewer extreme values in this analysis. The upper threshold was four and the lower threshold was one, suggesting that the majority of nodes in the network can be connected by a relatively short path.

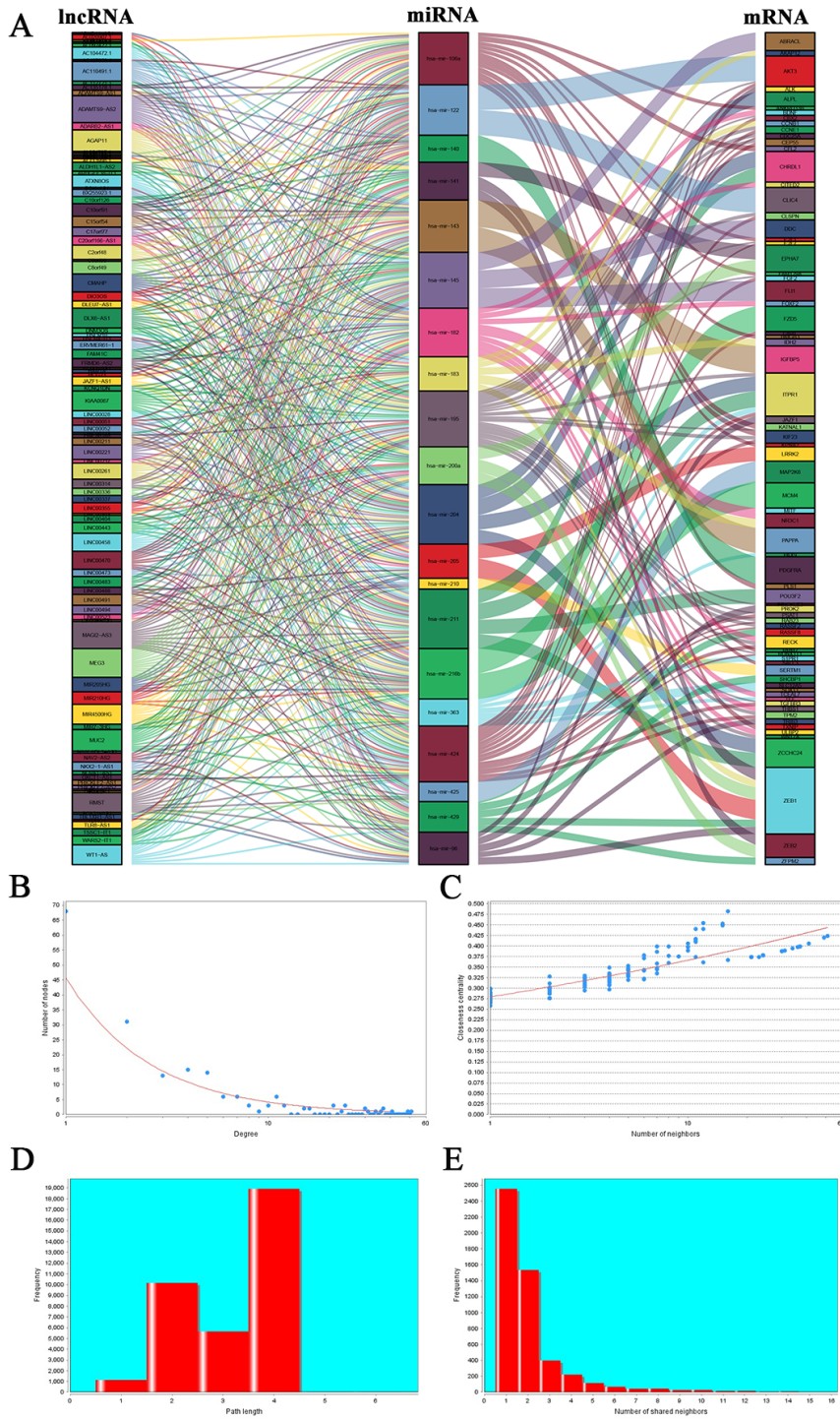

**Figure 2  Sankey diagram for the ceRNA network in EC and topology, and stability analysis.** (A) Each rectangle represents a gene, and the connection degree of each gene is displayed based on the size of the rectangle. (B) Node degree distribution analysis. (C) Closeness centrality distribution. (D) Shortest path distribution. (E) Node degree distribution density.

Node degree distribution density is shown in Fig. 2E. We found that the number of nodes decreased as node degree increased, indicating that most nodes in the network are isolated. It is possible that during diseases occurrence, some key nodes change and interact with nearby nodes; therefore, co-expression occur and downstream biological processes are affected. In addition, we calculated the connection degree of each gene by topology to clarify its importance in the ceRNA network (Table S5). MEG3 (connection degree = 21), hsa-mir-195 (connection degree = 51) and ZEB1 (connection degree = 6) were considered the most important genes among the lncRNAs, miRNAs, and mRNAs, respectively. Because hsa-mir-195 was the gene with the highest connection degree in the ceRNA network, we conclude that it might exert a strong influence on EC tumorigenesis.

## Positive correlations among the ceRNA expression levels

According to ceRNA theory, lncRNAs can positively regulate mRNAs by competitively competing with miRNAs (Fig. 2A). To verify this finding, regression analysis was performed between the DElncRNAs and DEmRNAs targeted by hsa-mir-195. Strong positive correlations were found between the DElncRNAs and DEmRNAs targeted by hsa-miR-195 (minimum requirement of an interaction score >0.3; Fig. 3; Table S6). Figures 3A–3F shows the top six interactions in which C2orf48 interacts with PSAT1, C2orf48 interacts with KIF23, C2orf48 interacts with CCNE1, C2orf48 interacts with CEP55, C2orf48 interacts with CBX2, and C2orf48 interacts with CDC25A under hsa-mir-195 regulation.

## Biological processes and pathway enrichment analysis of the ceRNA network

Seventy-three mRNAs in the ceRNA network were subjected to Gene Ontology (GO) and pathway enrichment analysis using the DAVID 6.7 database. The first 20 biological functions that demonstrated statistical significance were analyzed by "GOplot".R package. Pathways were visualized with Cytoscape software. GO analysis demonstrated that the biological processes significantly enriched by mRNAs in the ceRNA network include GO:0000187~activation of MAPK activity, GO:0060412~ventricular septum morphogenesis and GO:0045599~negative regulation of fat cell differentiation (Fig. 4A; Table S7). Kyoto Encyclopedia of Genes and Genomes (KEGG) pathway enrichment analysis demonstrated that these genes are involved in regulating microRNAs in cancer, p53 signaling pathways, the cell cycle, the Rap1 signaling pathway and other signaling pathways closely associated with tumorigenesis (Fig. 4B; Table S8).

## Survival and gene expression analysis

To clarify how DElncRNAs, DEmiRNAs, and DEmRNAs in the ceRNA network affect the prognoses of patients suffering from EC, K-M survival analysis was performed for the 97 lncRNAs, 20 miRNAs and 73 mRNAs, separately. The results demonstrate that 92 of the 97 lncRNAs were significantly associated with prognosis based on their respective optimal cutoffs (Kassambara et al., 2014) and 59 DElncRNAs exhibited positive correlations with overall survival; in contrast, the remaining 33 DElncRNAs exhibited negative associations with overall survival (log-rank $P < 0.05$; Table S9). Sixteen of the 20 differentially expressed miRNAs correlated to prognosis. Ten DEmiRNAs positively correlated to overall survival,

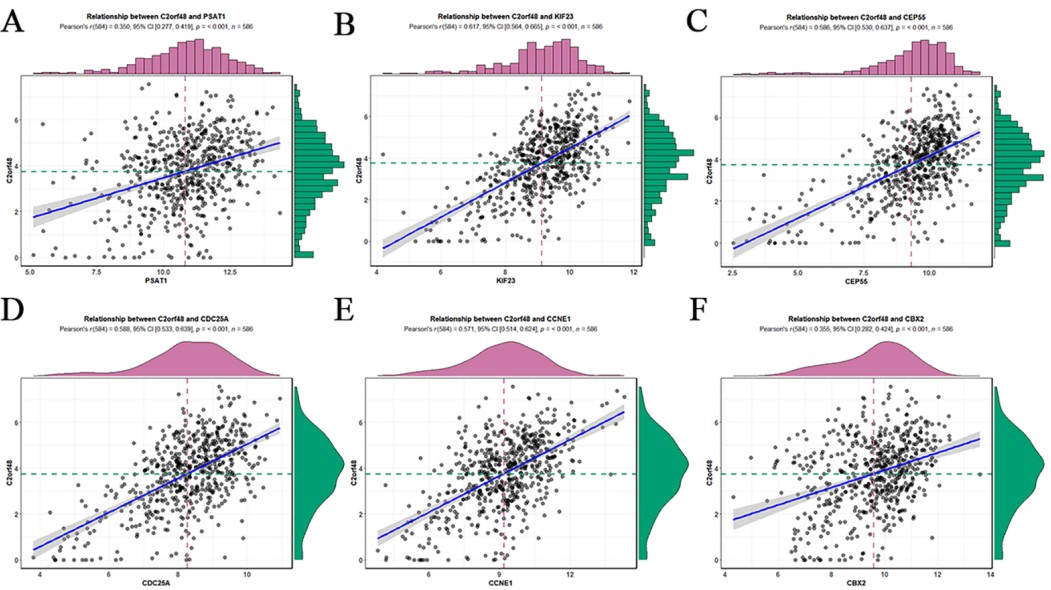

**Figure 3** **Regression analysis between the expression levels of DElncRNAs and DEmRNAs targeted by hsa-mir-195.** (A) Regression analysis between the expression levels of C2orf48 and PSAT1. (B) Regression analysis between the expression levels of C2orf48 and KIF23. (C) Regression analysis between the expression levels of C2orf48 andCEP55. (D) Regression analysis between the expression levels of C2orf48 and CDC25A. (E) Regression analysis between the expression levels of C2orf48 and CCNE1. (F) Regression analysis between the expression levels of C2orf48 and CBX2. The horizontal axis indicates mRNA expression, and the vertical axis indicates lncRNA expression. (Pearson's $r$, correlation coefficient.) The upper and right edges are histograms of gene expression.

and the remaining 6 DEmiRNAs were negatively associated with overall survival (log-rank $P < 0.05$; Table S10). Sixty-one of the 73 DEmRNAs were highly relevant for overall survival based on their respective optimal cutoffs. Forty-four DEmRNAs positively correlated to overall survival, and the remaining 17 DEmRNAs were negatively correlated to overall survival (log-rank $P < 0.05$; Table S11). Figures 5A–5D shows the survival curves of the top four DElncRNAs (FAM41C, ADARB2.AS1, C8orf49 and MIR7.3HG), Figs. 5E–5H shows the survival curves of the top four DEmiRNAs (has-mir-195, has-mir-140, has-mir-205 and has-mir-200a) and Figs. 5I–5L shows the survival curves of the top four DEmRNAs (PSAT1, KIF23, MCM4 and CCNE1). Then, we verified the expression of four mRNAs and miRNAs through the TCGA endometrial cancer dataset, and found that hsa-mir-195 and hsa-mir-140 are poorly expressed in endometrial cancer (Figs. 5Q–5R). The survival curves showed that as expression decreases, patient prognoses worsen, suggesting that these two miRNAs promote endometrial cancer development by negatively regulating the expression of cancer-promoting genes. Hsa-mir-205 and hsa-mir-200a were highly expressed in endometrial carcinoma tissues (Figs. 5S–5T), and their survival curves showed that as gene expression increases, survival prognosis worsens. We conclude that both miRNAs negatively regulate the expression of tumor suppressor genes and thereby promote the occurrence and development of tumors. The four genes that are highly expressed in endometrial cancer tissues (PSAT1, KIF23, CCNE1 and MCM4) showed a pattern
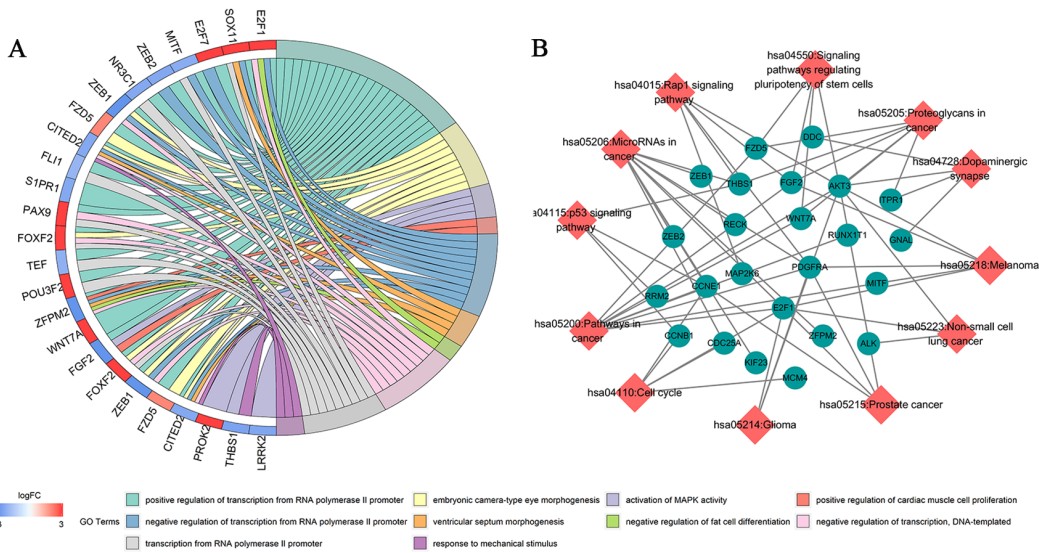

**Figure 4  Biological function and pathway enrichment analysis of 73 DEmRNAs.** (A) Chord diagram showing the top 10 enriched GO clusters for 73 DEmRNAs. In each chord diagram, enriched GO clusters are shown on the right and genes contributing to enrichment are shown on the left. Upregulated DEm-RNAs are displayed in red and downregulated DEmRNAs are displayed in blue. Each GO term is represented by one colored line. (B) Significant pathway enrichment of DEmRNAs. Diamonds represent the signaling pathways and ellipses represents the regulated genes.

whereby increasing expression was associated with worsening five-year survival prognosis (Figs. 5M–5P).

## Analysis of correlations between gene expression and clinicopathological parameters

The UALCA online database (*Chandrashekar et al., 2017*) was used to analyze correlations between mRNA expression of the above genes and patient clinicopathological parameters. We found that PSAT1, KIF23, CCNE1 and MCM4 expression was higher at different clinical stages than in the control group, and expression was higher in different pathological subtypes than in the normal group. At the same time, we found that the expression levels of all four genes was higher in the serous subtype than in endometroid type. All differences were statistically significant, suggesting that these four markers could also be used as biomarkers for molecular subtypes of EC (Figs. 6A–6H).

## Validation of DEmRNAs and DEmiRNAs in the ceRNA network

The top four DEmRNAs and DEmiRNAs correlated with overall survival were selected for validation. Consistent with our earlier results, mean expression levels of all four mRNAs were significantly higher in EC tissues than in non-cancerous tissues in the GSE17025 dataset (Figs. 7A–7D). Mean expression levels of hsa-mir-195, hsa-mir-140 were significantly lower in EC tissues than in non-cancerous tissues in the GSE35794 dataset (Figs. 7E–7F), and hsa-mir-205 expression was significantly higher in EC tissues.

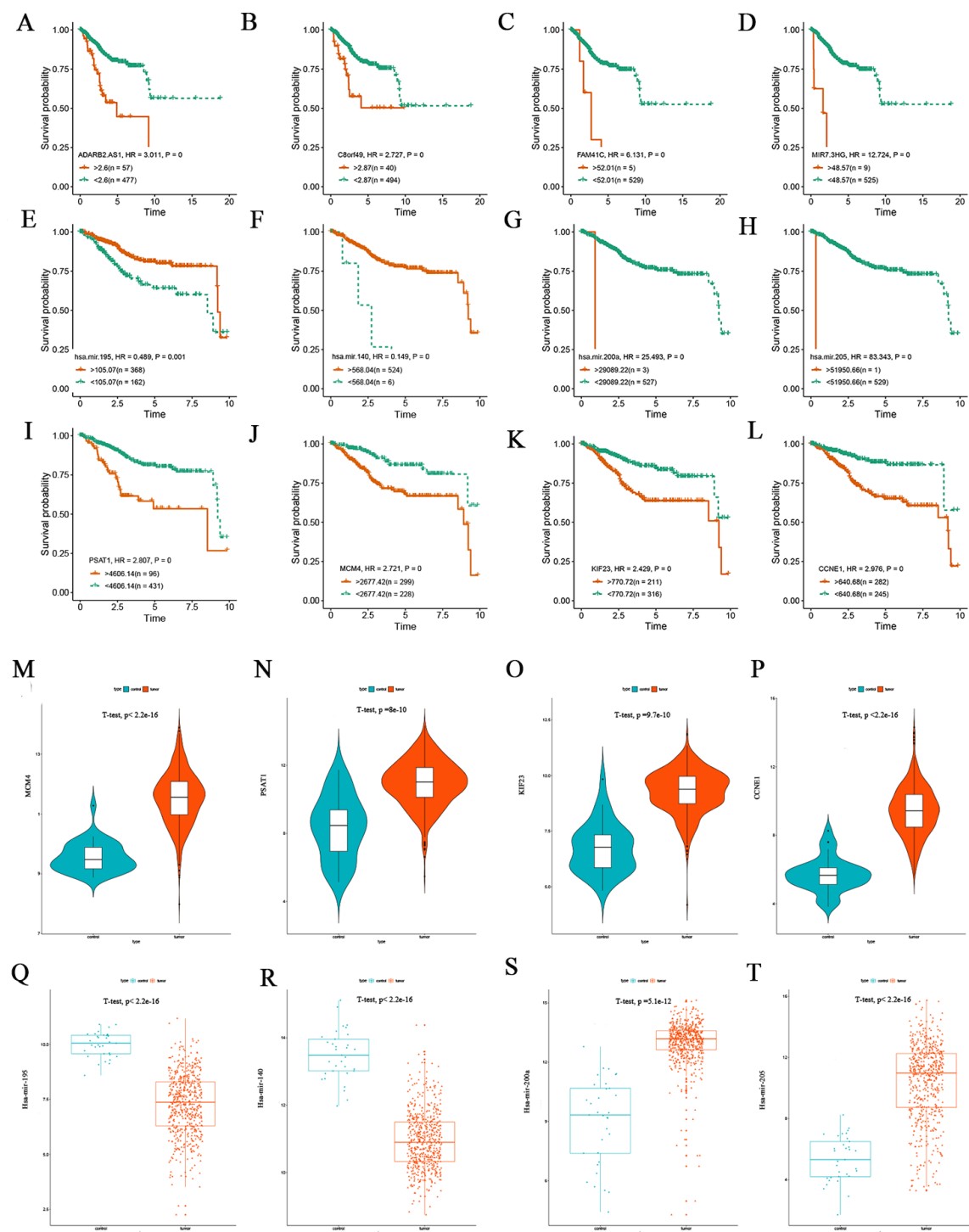

**Figure 5 Kaplan-Meier survival curves for the top four lncRNAs, miRNAs and mRNAs related to overall survival and expression validation.**
(A–D) Kaplan-Meier survival curves for the top four lncRNAs. (E–H) Kaplan-Meier survival curves for the top four miRNAs. (I–L) Kaplan-Meier survival curves for the top four mRNAs. (M–P) Expression validation of the top four DEmRNAs correlated with survival. (Q–T) Expression validation of the top four DEmiRNAs correlated with survival.

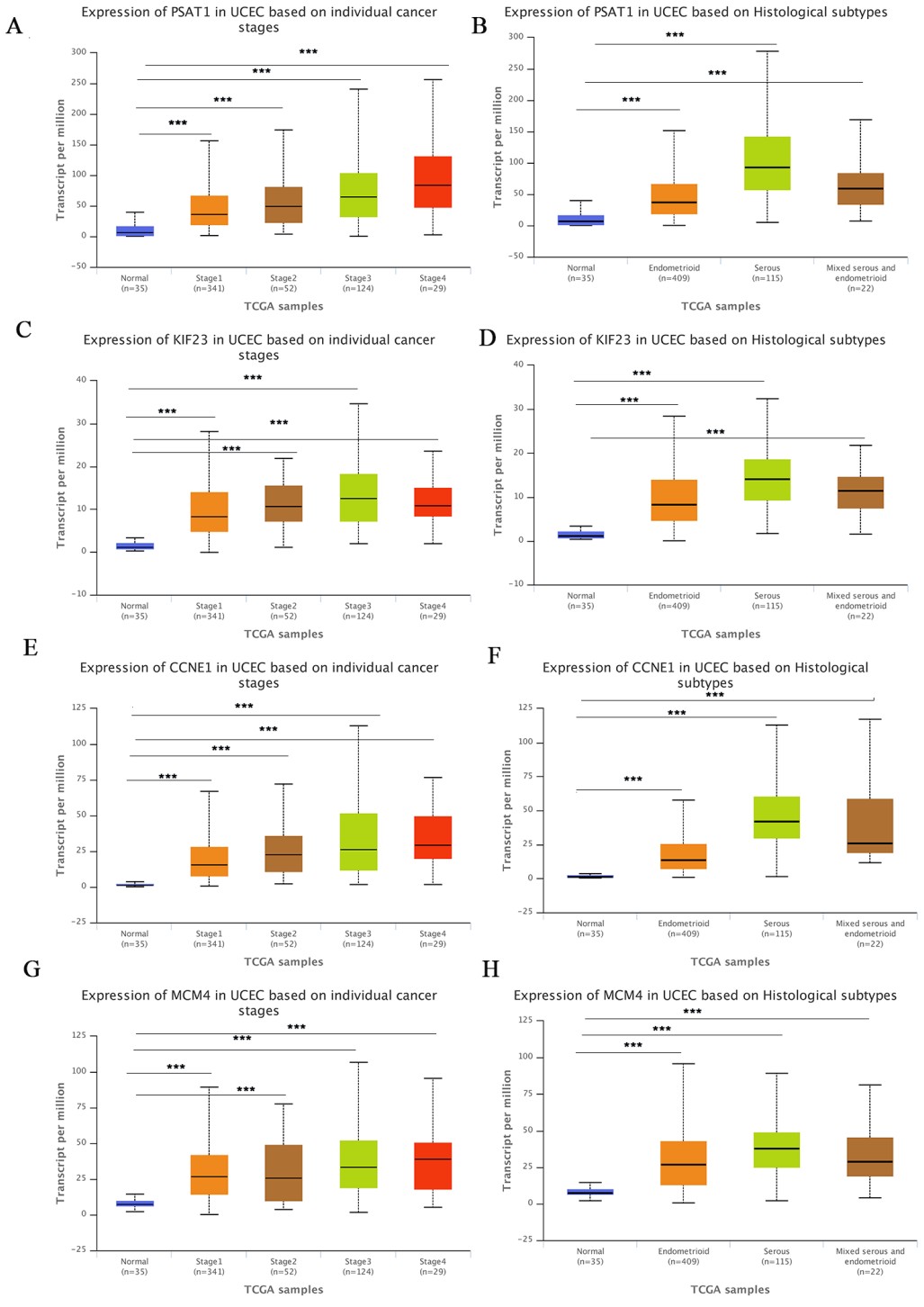

**Figure 6  Correlation between gene expression and clinicopathological parameters.** (A–B) Correlations between PSAT1 gene expression levels and clinical stage and histological classification in the TCGA database. (C–D) Correlations between KIF23 gene expression levels and clinical stage and histological classification. (E–F) Correlations between CCNE1 gene expression levels and clinical stage and histological classification. (G–H) Correlations between MCM4 gene expression levels and clinical stage and histological classification. ***$P$-value < 0.01.

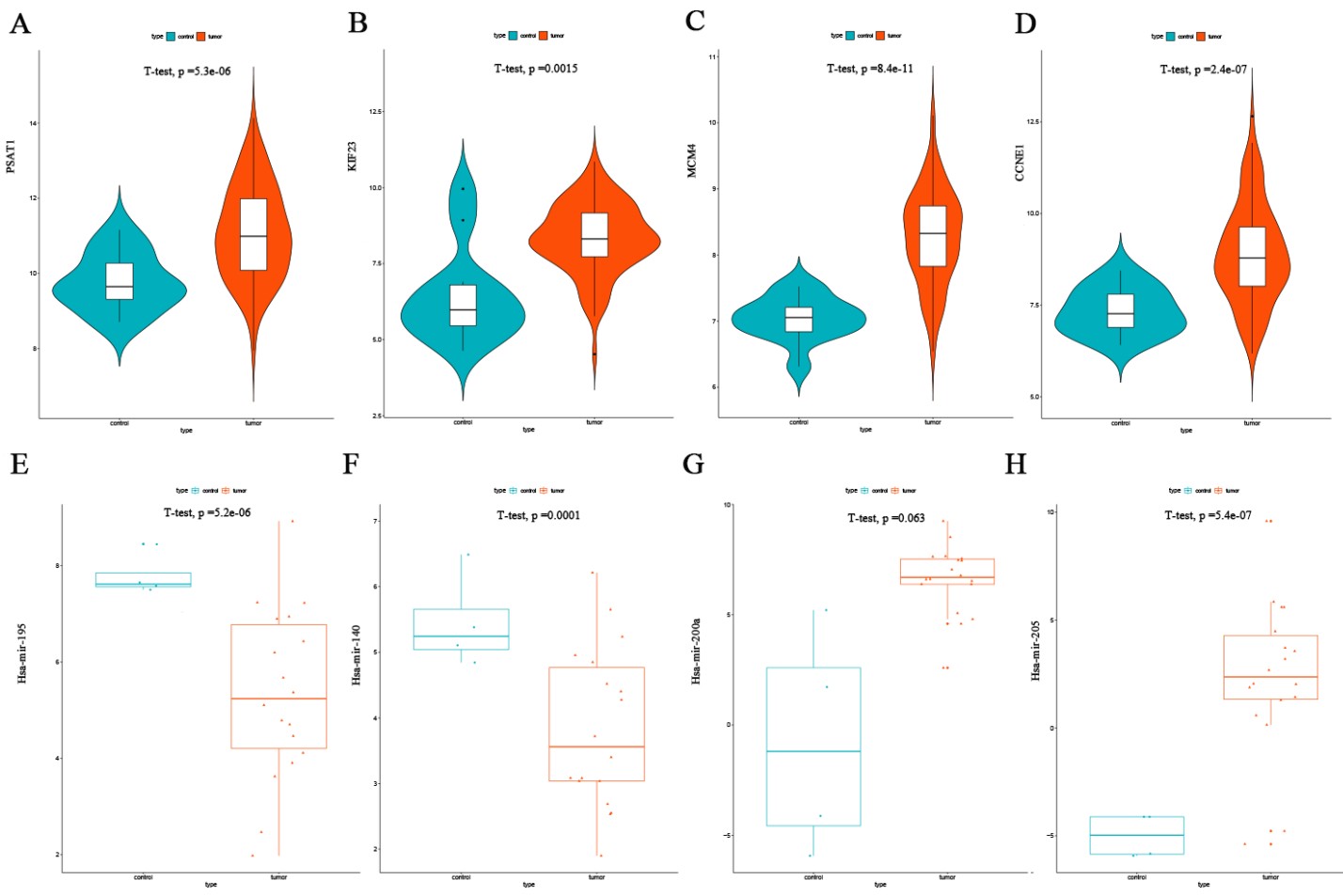

**Figure 7** **Validation of the top four DEmRNAs and DEmiRNAs correlated with survival.** (A) Violin plots of PSAT1mRNA expression levels in the GSE17025 validation dataset. (B) Violin plots of KIF23 mRNA expression levels in the GSE17025 validation dataset. (C) Violin plots of MCM4 mRNA expression levels in the GSE17025 validation dataset. (D) Violin plots of CCNE1 mRNA expression levels in the GSE17025 validation dataset. (E) Scatter plots of the has-mir-195 miRNA expression levels in the GSE35794 validation dataset. (F) Scatter plots of the has-mir-140 miRNA expression levels in the GSE35794 validation dataset. (G) Scatter plots of the hsa-mir-200a miRNA expression levels in the GSE35794 validation dataset. (H) Scatter plots of the hsa-mir-205 miRNA expression levels in the GSE35794 validation dataset.

However, due to the small number of patients, no significant differences were found in hsa-mir-200a expression (Figs. 7G–7H).

## Flow chart of construction and analysis of ceRNA network

The flow chart of construction of lncRNAs-miRNAs-mRNA regulation network in endometrial carcinoma of is shown in Fig. 8.

## DISCUSSION

Endometrial cancer (EC) is the most common gynecological tumor in women . Its carcinogenesis and progression are driven by multiple mechanisms, many of which interact with one another. Therefore, it is important for early diagnosis and treatment biomarker
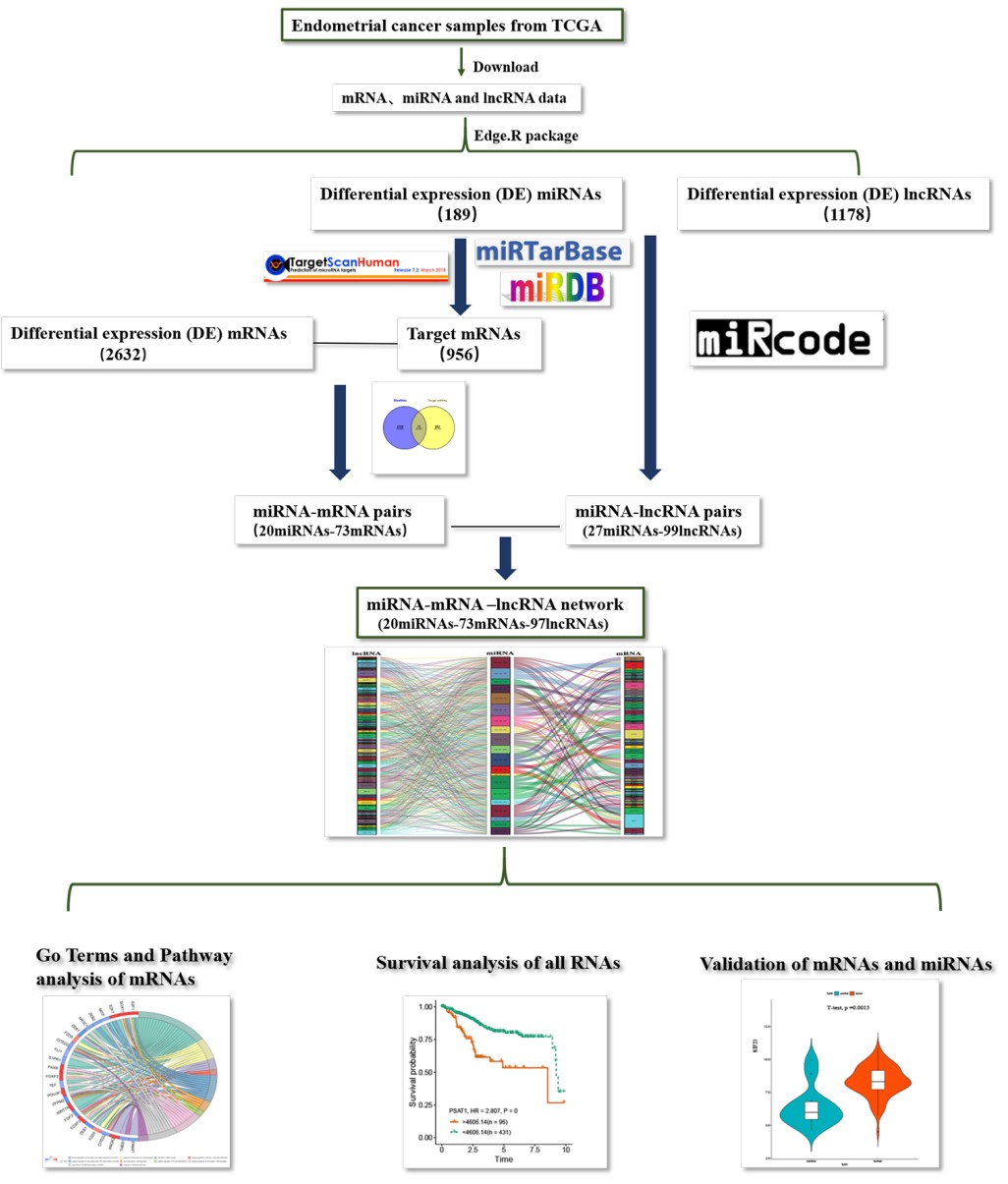

**Figure 8 Flow chart of construction and analysis of ceRNA network.** The flow chart of construction of lncRNAs-miRNAs-mRNA regulation network in endometrial carcinoma.

discovery to investigate the underlying mechanisms of EC occurrence and development, and to screen prognosis-associated RNAs.

A growing number of investigations have confirmed that lncRNAs act as key players regulating gene expression during the EC transformation and progression through signaling, decoying, guiding and scaffolding at the epigenetic, transcriptional and post-transcriptional levels. Many studies have indicated that a complicated regulatory network exists between lncRNAs and miRNAs. MicroRNAs (miRNAs) play important roles in post-transcriptional regulation by inhibiting target gene expression; EC carcinogenesis is also

associated with miRNA expression. Therefore, these RNAs could be used as cancer-related prognostic biomarkers.

In the present study, we constructed a ceRNA regulation network composed of 97 lncRNAs, 20 miRNAs and 73 mRNAs. Among these RNAs, the expression levels of 92 lncRNAs, 16miRNAs and 61 mRNAs were correlated to prognostic survival.

Genes related to overall survival are likely to reflect on the prognosis of EC.

lncRNAs are non-coding protein RNA molecules of more than 200 nucleotides in length, and recent studies have reported an increasing number of functional lncRNAs. Maternal expressed gene 3 (MEG3) is a maternal imprinted gene encoding a lncRNA that acts as a tumor suppressor in various tumors. In the constructed ceRNA topology network, lncRNA MEG3 showed the highest degree of connectivity of included DElncRNAs (degree of connectivity = 21), and its expression is downregulated in EC tissues. *Sun et al. (2017b)* found that MEG3 expression in EC tissues is significantly lower than in normal endometrial tissues. MEG3 overexpression inhibits proliferation, invasion and metastasis of EC cells and promotes apoptosis by inhibiting activation of the phosphoinositide 3-kinase (PI3K)/m-TOR signaling pathway. We also found that MEG3 is associated with 16 key DEmiRNAs (hsa-mir-429, hsa-mir-205, hsa-mir-141, hsa-mir-204, hsa-mir-182, hsa-mir-424, hsa-mir-140, hsa-mir-143, hsa-mir-216b, hsa-mir-106a, hsa-mir-96, hsa-mir-145, hsa-mir-200a, hsa-mir-122, hsa-mir- 211 and hsa-mir-195) and competes to regulate mRNA expression in EC. Of these, hsa-mir-195 is the highest of all nodes, and interacts with most of the DElncRNAs and DEmRNAs in the ceRNA network, suggesting that hsa-mir-195 has a significant effect on EC pathogenesis and prognosis. *Yang et al. (2014)* found that miR-195 is down-regulated in hepatocellular carcinoma tissues and cells. Overexpression of miR-195 can target Wnt3a, inhibit proliferation of hepatocellular carcinoma cells, and promote apoptosis. In our study, we found that miR195 is also significantly underexpressed in endometrial cancer tissues, and the survival curve suggests that its low expression is associated with poor prognosis. This indicates that miR195 inhibits the expression of oncogenes and promotes the development of endometrial cancer.

Our regression analysis of lncRNAs and mRNAs associated with miR195 revealed a significant correlation between C2orf48, PSAT1, KIF23, CCNE1, CDC25A and CBX2. Among these, C2orf48 had the strongest correlation with KIF23 (Person's $r = 0.617$; $p < 0.001$), and C2orf48, PSAT1, KIF23, CCNE1, CDC25A, and CBX2 were significantly upregulated in EC tissues as compared to adjacent tissues. Survival analysis showed that the higher the expression levels of lncRNAs C2orf48, PSAT1, KIF23, CCNE1, CDC25A and CBX2 in patients with endometrial cancer, the poorer the prognosis. *Furuta et al. (2013)* successfully demonstrated that miR195 can inhibit CDC25A and CCNE1 expression thereby regulating the cell cycle and inducing hepatoma cell proliferation.

To validate the expression of prognostic-related molecules in the constructed ceRNA network, we selected four mRNAs (PSAT1, KIF23, MCM4 and CCNE1) and miRNAs (mir-195, mir-140, mir-205 and mir-200a) that are the most relevant to prognosis in the ceRNA network, and verified in the mRNA and miRNA expression profiles in an external dataset of endometrial carcinoma from GEO. The results were consistent with those of the TCGA database. There was no significant difference in the differential expression of

mir-200a due to the small number of external samples. The above results demonstrate to some extent the reliability of the ceRNA network constructed by prognosis-related RNA after screening. At the same time, we used the TCGA database to analyze correlations between PSAT1, KIF23, CCNE1, MCM4 and clinical pathological parameters, finding that their expression levels in serous endometrial cancer are significantly higher than in endometriod endometrial carcinoma, suggesting that these four RNAS could also be used as molecular biomarkers of EC.

The KIF23 gene belongs to the KIF family and has been found to play an important role in the process of mitotic cytoplasmic separation. KIF23 overexpression is significantly associated with tumor grade, invasion and prognosis in breast cancer (*Zou et al., 2014*). High expression of KIF23 in glioma cells may be related to transcriptional activation, and *in vivo* and *in vitro* experiments have demonstrated.

KIF23 knockdown significantly inhibits glioma cell proliferation (*Takahashi et al., 2012*).

And it was also found to be abnormally expressed in gastric cancer, NSCLC tissues (*Murakami et al., 2013*; *Kato et al., 2016*).

However, there is no research report on KIF23 in EC. Our study found that it is highly expressed in EC, is regulated by mir-195 is associated with poor prognosis, and can also be used as a marker for a subtype of EC.

PSAT1 is an enzyme involved in serine synthesis. PSAT1 overexpression correlates to poor prognosis in colon cancer, non-small cell lung cancer and breast cancer (*Martens et al., 2005*; *Pollari et al., 2012*; *Vié et al., 2008*; *Yang et al., 2015*). To date, no report has focused on PSAT1 in EC. *Yan et al. (2015)* demonstrated that miRNA-340 inhibits esophageal cancer cell proliferation and invasion. PSAT1 overexpression is a marker for poor prognosis in nasopharyngeal carcinoma, as well (*Liao et al., 2016*). Based on our research, we believe that lncRNA C2orf48 can mediate PSAT1 expression by competing with miR-195, thereby promoting EC progression.

## CONCLUSION

To summarize, we constructed a ceRNA topological network based on RNA-seq and miRNA-seq data of EC in the TCGA public database, analyzed correlations of nodes in the network to patient survival, and identified multiple RNAs that possibly affect prognosis at the transcriptional level. We describe a C2orf48-hsa-mir-195 ceRNA functional model associated with EC development and prognosis, and verified prognostic RNA expression using external data.

Finally, it should be noted that we did not conduct clinical trials to verify the relationship between the predicted RNA molecules and EC. Clinical studies require appropriate sample sizes and longitudinal designs to verify the results: the purpose of this study is to identify potential prognostic value of lncRNAs, miRNAs and mRNAs by constructing ceRNA networks to serve as a basis for further experimental and clinical research. In addition, the data on normal and cancer samples in the TCGA database are not matched, and therefore our research results also need further validation via sample as well as *in vitro* and *in vivo* experiments.

### Funding

Funding was provided by the Shengjing Freelance Researcher Program (Program #201303). The funders had no role in study design, data collection and analysis, decision to publish, or preparation of the manuscript.

### Grant Disclosures

The following grant information was disclosed by the authors:
Shengjing Freelance Researcher Program: #201303.

### Competing Interests

The authors declare there are no competing interests.

### Author Contributions

- Ming-Jun Zheng conceived and designed the experiments, performed the experiments, analyzed the data, contributed reagents/materials/analysis tools, prepared figures and/or tables, authored or reviewed drafts of the paper, approved the final draft.
- Rui Gou and Wen-Chao Zhang authored or reviewed drafts of the paper, approved the final draft.
- Xin Nie, Jing Wang, Ling-Ling Gao, Juan-Juan Liu and Xiao Li approved the final draft.
- Bei Lin authored or reviewed drafts of the paper, approved the final draft, revise, proofread, provide ideas and guidance.

### Data Availability

ceRNA_code has been deposited at GitHub: https://github.com/mingjunzheng/ceRNA_code.

### Supplemental Information

Supplemental information for this article can be found online at http://dx.doi.org/10.7717/peerj.6091#supplemental-information.

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
