# Peer review of "Screening of prognostic biomarkers for endometrial carcinoma based on a ceRNA network"

_PeerJ, doi:10.7717/peerj.6091_

## Round 0.1 · original submission · Major Revisions

Your manuscript was reviewed by experts in the field and their comments were mostly favorable. However, the comments and questions raised by the reviewers need to be addressed prior to further consideration for publication

Reviewer 1 ·

Basic reporting

This work , although not pointing a novel idea , is basically well designed observational study.

the Introduction is basic but written plain and straight forward
The methods are well sectioned and summarised

Experimental design

The design is within scope . A basic observational study . The current knowledge regarding non coding RNA is vast and many studies relates to early diagnosis or prognosis

no technical or ethical issues are present

The methods are well described

Validity of the findings

Again , I do not find the result to be ground breaking .
The results are well described and discussed nd contributes to the body of knowledge in this field

Reviewer 2 ·

Basic reporting

No comment.

Experimental design

Functional and enrichment analysis of ceRNAs should be described in a more detailed manner in method section.
In general, methods should be described with sufficient information to be reproducible by another investigator.

Validity of the findings

Findings are awesome and discussion includes all the critical assessments of the findings with similar studies.

Reviewer 3 ·

Basic reporting

Needs work. I believe that the article uses clear English to describe the study design and setup but could use some work on the following aspects
1) There are a lot of spelling and grammatical errors that can be resolved with a spell check and some minor proofreading.
2) Additionally, the authors use confusing terminology which reduces the readability of the manuscript. For example, in lines 67-69 they say” We recruited 461 EC patients …” and follow it up by saying no ethics approval was required since they downloaded the data. This reduces clarity and occurs a few more times in the manuscript.
3) The results section is very hard to follow because the authors frequently describe the sub parts out of order (see explanations for figure 9, and 10) for example. This is compounded by the fact that most of the figure captions do not contain enough details. I encourage the authors to separately add a legend to each subpart of the figure in addition to the caption to improve the readability of the figures.
Additionally, text in the results section itself is often just 1-2 lines for the entire figure(See fig 5 and 6). I request the authors to invest some time in detailing the analysis, the findings and its significance to the biological question at hand for each figure. This will greatly help the reader understand the authors reasoning for performing the analysis
I believe that the background/context of the study has been laid out sufficiently and relevant literature has been cited. Although some more work is required on a few lines in the introduction (see lines 33-36 specifically), it is clear what the authors have set out to accomplish.
With respect to the figures, it’s my opinion that the authors have created some extraneous figures. For example, in figure 3, parts C and D while interesting, are redundant and do not really convey any biological relevance by themselves. The authors do not spend any of the discussion explaining their rationale behind the analysis strategy either and the reader could use some help following the scientific rationale for this analysis (this is especially true in case of figure 4 where they have applied network analysis and stated the results without any analyses of the results, why it was done in the first place or explaining why its significance).
Please try to focus the data in service of the question at hand, which in itself needs some clarification ( moving some figures to supplementary data will also help)
The submission seems self-contained

Experimental design

I believe that the current work is original primary research that is within the aims and scope of the journal.
I commend the authors for clearly defining the research question at hand and explaining its relevance to the understanding of RNA networks in EC
I believe that the authors have attempted to clearly explain the methodology. In order to encourage reproducibility of their analysis, the authors may consider hosting the code on a repository such as github so that all readers can access and recreate their findings.
I believe that the authors have thoroughly investigated the mRNA, lncRNA and miRNA profiles of endometrial carcinoma that was accessible from TCGA. I believe that the study meets the prevailing ethical standards in the field

Validity of the findings

I believe that the findings in this study are valid. However, especially for the part of the paper that deals with generating the prognostic markers, I believe that the authors need to separate the data into a training and validation data set. They have data from 463 patients and I believe that they can easily accomplish this. This will greatly enhance the credibility of the generated biomarkers instead of solely relying on the AUROC number and the Kaplan Meier curves.
I find the manuscript somewhat hard to follow since the authors identify 8 prognostic markers, calculate their predictive power and build a network around them only to identify two more mRNA( PSAT1and KIF23)? If these are the two genes they are proposing as biomarkers, can they rerun the AUROC with them( for a total of 10 mRNA or use just these 2) ? If not, why was this part of the analysis performed? It seems removed from the original intent of the paper and seems to take it in an unclear direction. I am also confused by them running GSEA on these 2 genes and analyzing it instead of analyzing the GSEA profiles of the 8 mRNA that they propose. I think that this part of the paper merits more explanation of scientific rationale.

Additional comments

I believe that the authors have made a strong attempt to understand the interactions between mRNA-miRNA and lnRNA expression in endometrial cancer. Their analysis while often excessive cannot be faulted for not being thorough.
Overall, I think that the readability and applicability of the paper will be greatly improved by focusing
1) The questions that the authors are trying to address: are you identifying prognostic markers or performing a RNA network analysis?
2) The data you are providing to answer this question: some figures are redundant and others have not been explained thoroughly. Then sub-network analysis section of the paper in particular needs to be evaluated with this lens

---

## Round 0.2 · Minor Revisions

The manuscript was substantially revised and improved.

One of the reviewers raised minors issues that need to be addressed prior to its acceptance for publication

Reviewer 3 ·

Basic reporting

I believe that the article uses clear English to describe the study design and setup
I believe that the background/context of the study has been laid out sufficiently and relevant literature has been cited.
The submission is self contained and the figures make sens

Experimental design

I believe that the current work is within the aims and scope of the journal.
I believe that the authors have attempted to clearly explain the methodology. In order to encourage reproducibility of their analysis, the authors may consider hosting the code on a repository such as github so that all readers can access and recreate their findings.
I believe that the authors have thoroughly investigated the mRNA, lncRNA and miRNA profiles of endometrial carcinoma that was accessible from TCGA. I believe that the study meets the prevailing ethical standards in the field

Validity of the findings

I believe that the findings in this study are valid.

Additional comments

I believe that the authors have addressed most of my concerns and re-worked the paper sufficiently. I commend the authors on reworking the figures to present a coherent paper.
There are some minor edits
1) Some figures are not described in the text still (eg 5K and 5L).
2) In order to encourage reproducibility of their analysis, the authors may consider hosting the code on a repository such as github so that all readers can access and recreate their findings.
3) i don't follow how the authors validated their top 4 miRNA and mRNA's since they mention they didnt do a training and validation data set? How was figure 7 generated? please clarify: Is this qPCR data?
4) the discussion is lengthy and gives too may examples of RNA's and their roles in cancer. Some editing will help contextualize this study better
5) it gets a little confusing with all the lincRNA, mRNA and miRNA names being used int he manuscript. is it possible to make a simplified schematic summarizing the findings at the very end?

---

## Round 0.3 · accepted · Accept

The authors addressed the comments of the reviewer and the manuscript can be accepted for publication

#